# The development of a novel diagnostic PCR for *Madurella mycetomatis* using a comparative genome approach

Wilson Lim[1], Emmanuel Siddig[1,2], Kimberly Eadie[1], Bertrand Nyuykonge[1], Sarah Ahmed[3¤a¤b], Ahmed Fahal[2], Annelies Verbon[1], Sandra Smit[4], Wendy WJ van de Sande[1]*

**1** Erasmus MC, University Medical Center Rotterdam, Department of Microbiology and Infectious Diseases, Rotterdam, The Netherlands, **2** Mycetoma Research Centre, University of Khartoum, Khartoum, Sudan, **3** Westerdijk Fungal Biodiversity Institute, Utrecht, The Netherlands, **4** Wageningen University & Research, Department of Plant Science, Wageningen, The Netherlands

¤a  Current address: Center of Expertise in Mycology of Radboud University Medical Center / Canisius Wilhelmina Hospital, Nijmegen, The Netherlands
¤b  Current address: Faculty of medical laboratory sciences, University of Khartoum, Sudan
* w.vandesande@erasmusmc.nl

**Data Availability Statement:** All relevant data are within the manuscript.

**Funding:** The authors received no specific funding for this work.

## Abstract

### Background

Eumycetoma is a neglected tropical disease most commonly caused by the fungus *Madurella mycetomatis*. Identification of eumycetoma causative agents can only be reliably performed by molecular identification, most commonly by species-specific PCR. The current *M. mycetomatis* specific PCR primers were recently discovered to cross-react with *Madurella pseudomycetomatis*. Here, we used a comparative genome approach to develop a new *M. mycetomatis* specific PCR for species identification.

### Methodology

Predicted-protein coding sequences unique to *M. mycetomatis* were first identified in BLASTCLUST based on E-value, size and presence of orthologues. Primers were then developed for 16 unique sequences and evaluated against 60 *M. mycetomatis* isolates and other eumycetoma causing agents including the *Madurella* sibling species. Out of the 16, only one was found to be specific to *M. mycetomatis*.

### Conclusion

We have discovered a predicted-protein coding sequence unique to *M. mycetomatis* and have developed a new species-specific PCR to be used as a novel diagnostic marker for *M. mycetomatis*.

**Competing interests:** The authors have declared that no competing interests exist.

## Author summary

Mycetoma is a neglected tropical disease characterised by tumorous swellings and grain formation. This disease can be caused by more than 70 different micro-organisms and is categorised into actinomycetoma (caused by bacteria) and eumycetoma (caused by fungi). The most common causative agent of mycetoma is the fungus *Madurella mycetomatis*. Diagnosis of eumycetoma is often only done clinically or by histopathological examination and culturing of the grains. Unfortunately, that often leads to misidentifications. Molecular identification is currently the most reliable method to identify the causative agents. However, we have recently discovered that the only *M. mycetomatis* species-specific PCR primers cross-reacts to *Madurella pseudomycetomatis*. Since all *Madurella* species cause eumycetoma and have different susceptibilities to antifungal agents, it is important to be able to accurately identify them to the species level. Here we have used a comparative genome approach to identify and design new *M. mycetomatis* species-specific PCR primers. These primers can be used to identify *M. mycetomatis* directly from grains and do not cross-react with any of the other eumycetoma causative agents tested. We, therefore, recommended the use of these primers in reference centres and local laboratories to identify *M. mycetomatis* to the species level.

## Introduction

The neglected tropical disease mycetoma presents itself as a subcutaneous chronic granulomatous inflammatory disease and is characterized by tumorous lesions and grain formation [1,2]. This disease can be caused by more than 70 different micro-organisms and is categorized into actinomycetoma (caused by bacteria) and eumycetoma (caused by fungi). Treatment is dependent on the causative agent.

Diagnosis of mycetoma is often only made clinically in endemic areas due to the scarcity of facilities, expertise and financial capacity. Eumycetoma and actinomycetoma can be easily distinguished from each other by the texture and colour of the grains. However, species identification based on the texture and colour of the grain is not possible as many fungal species can produce similar-looking black-grains. Therefore, identification of causative agents is usually done by histopathological examination and culturing of grains. Unfortunately, that often leads to misidentifications [3]. Currently, molecular identification is the most reliable method to identify eumycetoma causative agents down to the species level and the most commonly used technique is amplifying the Internally Transcribed Spacer (ITS) region and sequencing [4]. However, many endemic areas lack the ability to perform DNA sequencing.

In 1999, a species-specific PCR primer based on the internal transcribed spacer (ITS) was developed for *Madurella mycetomatis*, the most common causative agent of mycetoma [5]. This PCR is currently used at the Mycetoma Reference Center in Khartoum, Sudan. It is performed on DNA obtained from cultured clinical material or directly from grains obtained from patients. Unfortunately, this *M. mycetomatis* specific PCR primer pair 26.1a and 28.3a was only recently discovered to cross-react with *Madurella pseudomycetomatis* [6]. *Madurella pseudomycetomatis* was not yet described at the time when the *M. mycetomatis* specific PCR was developed [7]. All four *Madurella* species (*Madurella fahalii*, *Madurella tropicana*, *M. mycetomatis* and *M. pseudomycetomatis*) are known to cause black grain eumycetoma and have different susceptibilities to antifungal agents. For instance, *M. fahalii* is not inhibited by itraconazole *in vitro*, which could have consequences for treatment strategy [8]. This makes identification of causative agents to the species level a must. Since all four *Madurella* species

share a very conserved ITS region, this has made designing PCR primers specific for *M. mycetomatis* based on that region difficult [8,9]. To circumvent this difficulty, we took a comparative genome approach to design a new set of specific primers for the identification of *M. mycetomatis* by PCR.

Here, we report a new set of diagnostic DNA primers for *M. mycetomatis* identified from the genome of *M. mycetomatis* [10].

## Materials and methods

### Ethics statement

This study was approved by the Mycetoma Research Center, Khartoum, Sudan (IRB, No. 11/2018). Written informed consent was obtained from each adult patient, and assent was taken from minors (aged below 18 years) with written consent from their guardian.

### Fungal isolates, patient grains and DNA isolation

A total of 95 fungal isolates were used in this study; 60 *M. mycetomatis*, 4 *M. tropicana*, 3 *M. fahalii*, 3 *M. pseudomycetomatis*, 1 *Aspergillus fumigatus*, 1 *Aspergillus terreus*, 2 *Chaetomium globosum*, 4 *Falciformispora senegalensis*, 1 *Fusarium solani*, 3 *Medicopsis romeroi*, 2 *Scedosporium apiospermum*, 3 *Thielavia terrestris*, 3 *Thielavia subthermophilia*, 4 *Trematospheria grisea* and 1 *Trichophyton rubrum*. Most fungal isolates were obtained from both the Mycetoma Research Center in Sudan and the Westerdijk Fungal Biodiversity Institute in the Netherlands and maintained in Erasmus Medical Centre. All *M. mycetomatis* isolates originated from mycetoma patients. Isolates are maintained on Sabouraud Dextrose (SAB) agar at either 37˚C or room temperature depending on the fungal species. Black grains were obtained from a total of 16 eumycetoma infected patients seen at the Mycetoma Research Center in Sudan. Nine patients were confirmed to be infected with *M. mycetomatis*, four with *F. senegalensis*, and three with *F. tompkinsii*. DNA from fungal isolates and grains were isolated as described earlier using the ZR Fungal/Bacterial DNA MicroPrep kit (Zymo Research, USA) [11]. All isolates were identified to the species level based on morphology, polymerase chain reaction (PCR)-based restriction fragment length polymorphisms, and sequencing of the ITS regions [5,12,13].

### Identifying specific predicted protein-coding sequences to *M. mycetomatis*

Predicted protein-coding sequences (PPCS) of *M. mycetomatis* were obtained from the published genome sequence of *M. mycetomatis* isolate mm55, accession number LCTW00000000, BioProject PRJNA267680 [10]. To determine their specificity to *M. mycetomatis*, a bioinformatical comparison of these sequences to the genome of other organisms was performed using BLASTCLUST [14]. The specificity of these PPCS were determined based on presence of orthologues, E-value and fragment size. Orthologues were defined as sequences with greater than 85% amino acid similarity to the tested *M. mycetomatis* PPCS. *M. mycetomatis* PPCS with no orthologues present in the genomes of other organisms, E- value of 0.003 and higher and size between 400 bp and 1100 bp were considered to be specific to *M. mycetomatis* and were chosen for further analysis.

### Primer design and PCR conditions

Forward and reverse PCR primers were designed according to the nucleotide sequence of the PPCS of interest. Primer sequences are depicted in Table 1. PCR reaction contained 0.6 units of Super Taq HC DNA polymerase (Sphaero Q), 0.1 nM/μl DNTP (Thermo Fisher Scientific) and

**Table 1. The sixteen predicted protein sequences with their corresponding size, primer sequences and annealing temperatures.**

| Primer set | Sequence length (bp) | E value | Primers (5'-3') | | Annealing temperature ˚C |
|---|---|---|---|---|---|
| 1 | 972 | 740 | F | ATGCCTGCCCGGTCAGTTCG | 55 |
| | | | R | CTAGTACATGCCCACAACCG | |
| 2 | 832 | 96 | F | ATGCGCTTTCTCTCCCTTAC | 55 |
| | | | R | TCAGCACTCCCTGATCAACC | |
| 3 | 808 | 35 | F | ATGCTGCTCGAAAGGGTGTC | 55 |
| | | | R | TCAACCCCGCCCCGTACCCG | |
| 4 | 639 | 0.006 | F | ATGCACTTCTTCAACACTGT | 55 |
| | | | R | CTAGACGGAGACACCTAGGG | |
| 5 | 636 | 1.3 | F | ATGAAGCTCACTGTCTCCCT | 55 |
| | | | R | TCAAAGAACAAAAGAGGCAG | |
| 6 | 621 | 1.9 | F | ATGAAGTACTCTAGCACTCT | 55 |
| | | | R | TTAGGCCGCCTGGGTGGCCG | |
| 7 | 564 | - | F | ATGAAGCTCATCTCCATCGT | 55 |
| | | | R | TCACAAGAGGTACACAACAG | |
| 8 | 561 | 0.28 | F | ATGCAGCTCTCGATCGCCAA | 55 |
| | | | R | TTAAAGCAACATAGCCGCGT | |
| 9 | 677 | 2.1 | F | ATGGATCGCCTCGTCAAACC | 55 |
| | | | R | CTAAGTCAACAGAACGACAG | |
| 10 | 639 | 2.5 | F | ATGAGGTGGCTCGAGACGAC | 55 |
| | | | R | CTATGGTTGTCCACACCCAT | |
| Mmy-Fw Mmy-Rv | 474 | 20 | F | TCTCCTGTCCTACGACATCTGTGG | 59 |
| | | | R | TTCCTCACCTCCCAGCCCTTT | |
| 12 | 1089 | 0.007 | F | ATGGTGGAGCAGCTCTTGGT | 55 |
| | | | R | TCAAGGAATCGTTCTCGTAA | |
| 13 | 852 | 22 | F | ATGCATCAACGACATCTTGC | 55 |
| | | | R | CTAGAATTCCTGACGAGAAA | |
| 14 | 504 | 42 | F | ATGAAATTCACGGACTCTGG | 55 |
| | | | R | CTACATCAGCGGGCACTCCT | |
| 15 | 544 | 5.8 | F | ATGACAATCACAATCACAAT | 55 |
| | | | R | AAGCTGGCCCCCGATCACAG | |
| 16 | 544 | 0.91 | F | AGTAATCTAGTCACAATGGC | 55 |
| | | | R | TCAACCCGTGAAAATATTGC | |
| *26.1A *28.3A | 420 | - | F | AATGAGTTGGGCTTTAACGG | 58 |
| | | | R | TCCCGGTAGTGTAGTGTCCCT | |
| *26.1B *28.3B | 360 | - | F | GCAACACGCCCTGGGCGA | 58 |
| | | | R | TCCGCGGGGCGTCCGCCGGA | |

*M. mycetomatis* specific primers designed in 1999 [5].

0.5 pmol/µl of each forward and reverse primer. PCR conditions were as follows: initial denaturation at 94˚C for 10 min; 40 cycles of amplification with various annealing temperatures (95˚C for 1 minute, 55–59˚C for 1 minute, and 72˚C for 1 minute); and a final extension step of 10 seconds at 72˚C. The PCR reaction products were visualized in 2% agarose gel (Sphaero Q).

## Results and discussion

Since we have demonstrated that the currently used *M. mycetomatis* specific PCR cross-reacted with *M. pseudomycetomatis*, there was a need to develop a new *M. mycetomatis* specific PCR for

proper species identification. From the genome of *M. mycetomatis*, 350 predicted protein-coding sequences (PPCS) were randomly selected and analysed. We chose to analyse PPCS because these protein-coding sequences are likely to be more stable than non-coding sequences [15,16]. To ensure that they can be easily amplified through PCR, we preferentially chose PPCS with sizes between 400 and 1100 bp. From the initial 350 PPCS, the top 16 candidates that fitted our requirement based on specificity and size were chosen for PCR development.

PCR primers for the 16 candidates were then designed (Table 1). To ascertain that these primers would amplify their targets in all *M. mycetomatis* isolates, they were evaluated in 60 *M. mycetomatis* isolates from different geographical origins, genotypic backgrounds and phenotypic appearance. Out of the 16 primer sets tested, 13 were positive in all 60 *M. mycetomatis* isolates tested (Fig 1). Primer sets 4, 5 and 12 were present in 58, 4 and 59 isolates, respectively (Fig 1). To determine the specificity of the 13 positive primer sets to *M. mycetomatis*, they were tested against other fungal mycetoma causative agents and close relatives of *M. mycetomatis*. As seen in Table 2, only primer set 11 –later renamed as Mmy-Fw and Mmy-Rv—was found to be specific for *M. mycetomatis*. Primer set 2, 4, 8 and 9 were not able to discriminate between the different *Madurella* species while 5 and 7 could discriminate between the four *Madurella* species but cross-reacted with at least one other mycetoma causative agent. The amplicon generated by Mmy-Fw and Mmy-Rv appears to be a putative single-copy gene. To determine if these PCR primers were as sensitive as the currently used ones, we compared the two PCRs head-on. Mmy-Fw and Mmy-Rv were able to detect DNA concentrations as low as 5 pg. This is only slightly less sensitive compared to the currently used diagnostic PCR primer pair 26.1a and 28.3a that is able to detect DNA at 0.5 pg.

Primers Mmy-Fw and Mmy-Rv were also tested on DNA extracted from grains obtained from eumycetoma patients. As shown in Fig 2, amplicons were only observed when DNA

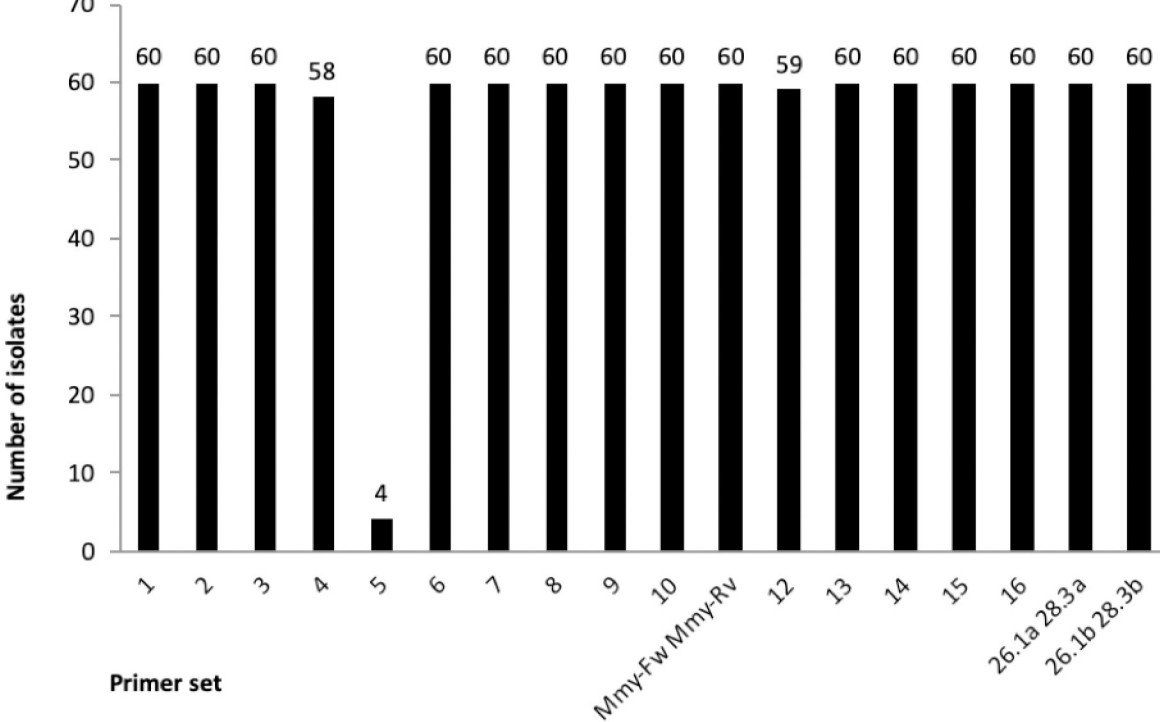

**Fig 1. Presence of the 16 PCR amplicons in 60 *M.** mycetomatis* isolates tested. Most PCR reactions resulted in amplification in all isolates tested except PCR 4, 5 and 12.

**Table 2. Presence or absence of PCR amplicons of the sixteen primer sets and PCR primers developed in 1999 [5] in the other eumycetoma causing agents and close relatives of *M. mycetomatis*.** No amplicons were observed in all species tested here using Mmy-Fw and Mmy-Rv. Only PCR with bands of the same sizes to *M. mycetomatis* is considered specific to *M. mycetomatis*.

| Primer set | 1 | 2 | 3 | 4 | 5 | 6 | 7 | 8 | 9 | 10 | Mmy-Fw Mmy-Rv | 12 | 13 | 14 | 15 | 16 | *26.1A 28.3A | *26.1B 28.3B |
|---|---|---|---|---|---|---|---|---|---|---|---|---|---|---|---|---|---|---|
| *Madurella tropicana* (4) | A | A | C | A | C | A | C | A | A | A | C | A | C | B | B | C | C | A |
| *Madurella fahalii* (3) | C | A | C | A | C | C | C | A | A | A | C | C | C | B | B | B | C | A |
| *Madurella pseudomycetomatis* (3) | A | A | A | A | C | B | C | A | A | B | C | B | B | C | B | A | B | A |
| *Aspergillus fumigatus* (1) | - | - | - | - | - | - | C | - | - | - | C | - | - | - | - | - | - | - |
| *Aspergillus terreus* (1) | - | - | - | - | - | - | C | - | - | - | C | - | - | - | - | - | - | - |
| *Chaetomium globosum* (2) | - | - | - | - | - | - | B | - | - | - | C | - | - | - | - | - | - | - |
| *Falciformispora senegalensis* (4) | B | B | B | B | B | B | C | B | B | B | C | C | B | B | C | B | C | C |
| *Fusarium solani* (1) | C | C | B | C | B | B | C | B | B | A | C | C | B | B | B | B | C | C |
| *Medicopsis romeroi* (3) | - | - | - | - | - | - | A | - | - | - | C | - | - | - | - | - | C | C |
| *Scedosporium apiospermum* (2) | - | - | - | - | - | - | - | - | - | - | C | - | - | - | - | - | C | - |
| *Thielavia subthermophilia* (3) | B | C | B | C | C | B | C | B | B | B | C | B | B | B | B | C | C | C |
| *Thielavia terrestris* (3) | B | B | B | C | C | B | C | B | B | B | C | B | B | B | B | C | C | C |
| *Trematosphaeria grisea* (4) | - | - | - | - | - | - | C | - | - | - | C | - | - | - | - | - | C | C |
| *Trichophyton rubrum* (1) | - | - | - | - | - | - | C | - | - | - | C | - | - | - | - | - | C | C |

A: PCR band of the same size; B: PCR band of another size; C: no PCR band.

*M. mycetomatis* specific primers designed in 1999 [5].

from *M. mycetomatis* grains were present. The primers did not cross-react to DNA obtained from *F. senegalensis* or *F. tompkinsii* grains (Fig 2). Our findings show that these primers are sufficiently sensitive to be used in diagnosis directly from clinical specimens.

One of the advantages of using this comparative genome approach is that primer designs are less constrained since the targeted genes are unique. With this method, we were able to design primers that can distinguish between *M. mycetomatis* and *M. pseudomycetomatis*. Other studies have also succeeded in designing specific primers for their organism of choice

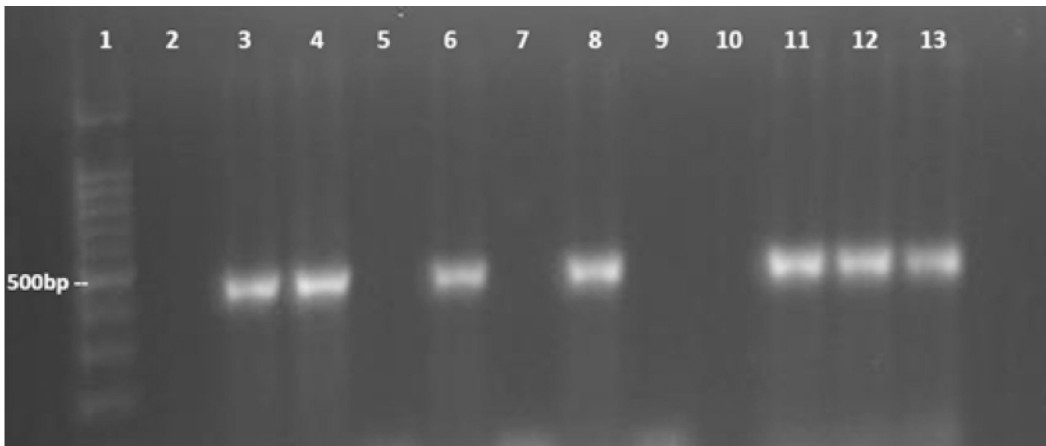

**Fig 2. The specificity of Mmy-Fw and Mmy-Rv on DNA isolated from eumycetoma grains.** Lane 1, 100 bp DNA ladder; Lane 2, negative control; Lane 3, 4, 6, 8, 11 and 12, *Madurella mycetomatis* DNA extracted from grains; Lane 5 and 7, *Falciformispora senegalensis* DNA extracted from grains; Lane 9 and 10, *Falciformispora tompkinsii* DNA extracted from grains; Lane 13, *Madurella mycetomatis* DNA from isolate as a positive control. Presence of amplicons on lane 3, 4, 5, 6, 8, 11 and 12 and none on the other lanes confirms the specificity of Mmy-Fw and Mmy-Rv towards *M. mycetomatis*.

using this approach [17–19]. In a study by Withers *et al*, a similar genome comparison method was performed on *Pseudoperonospora cubensis* and *Pseudoperonospora humuli* [19]. The comparison was first performed *in silico* and subsequently *in vitro*. Using this approach, they were able to identify and determine a large number of specific markers for their organism of interest while reducing the number of diagnostic candidates to validate with PCR [19]. However, a similar *in silico* approach could not be performed in our study because at the time of data analysis and the preparation of this manuscript, only the genome of one *M. mycetomatis* isolate and none of *M. fahalii*, *M. tropicana* and *M. pseudomycetomatis* was sequenced.

In conclusion, since cross-reactivity occurs with the current *M. mycetomatis* specific PCR primer pair 26.1a and 28.3a, we have used a comparative genome approach to identify and designed new *M. mycetomatis* species-specific PCR primers. Since new fungi causing eumycetoma are still being discovered, proper identification of its causative agents can help to fully understand the epidemiology and global burden of this disease. Thus, there is clearly a need for a specific PCR marker to identify its causative agents. We recommend reference centers such as the WHO collaborative Mycetoma Reference Center in Khartoum, Sudan and yet to be established reference laboratories in other endemic countries to use the new PCR primers Mmy-Fw and Mmy-Rv to identify *M. mycetomatis* to the species level. Furthermore, this comparative genome approach may also be used to design markers for other eumycetoma agents and also other fungi that share conserve ITS region within its genus.

## Author Contributions

**Conceptualization:** Wendy WJ van de Sande.

**Data curation:** Wilson Lim, Sandra Smit, Wendy WJ van de Sande.

**Formal analysis:** Wilson Lim.

**Funding acquisition:** Wendy WJ van de Sande.

**Investigation:** Wilson Lim, Kimberly Eadie.

**Methodology:** Wilson Lim, Emmanuel Siddig, Kimberly Eadie, Bertrand Nyuykonge.

**Project administration:** Wendy WJ van de Sande.

**Resources:** Wilson Lim, Sarah Ahmed, Ahmed Fahal, Sandra Smit, Wendy WJ van de Sande.

**Supervision:** Annelies Verbon, Wendy WJ van de Sande.

**Validation:** Wilson Lim.

**Visualization:** Wilson Lim.

**Writing – original draft:** Wilson Lim.

**Writing – review & editing:** Wilson Lim, Bertrand Nyuykonge, Sarah Ahmed, Ahmed Fahal, Annelies Verbon, Sandra Smit, Wendy WJ van de Sande.

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
