## [Decision Letter · Decision Letter 0]

4 Aug 2020

Dear Mr. Lim,

Thank you very much for submitting your manuscript "The development of a novel diagnostic PCR for Madurella mycetomatis using a comparative genome approach" for consideration at PLOS Neglected Tropical Diseases. As with all papers reviewed by the journal, your manuscript was reviewed by members of the editorial board and by several independent reviewers. The reviewers appreciated the attention to an important topic. Based on the reviews, we are likely to accept this manuscript for publication, providing that you modify the manuscript according to the review recommendations. 

Sincerely,

Husain Poonawala

Guest Editor

Todd Reynolds

Deputy Editor

Reviewer's Responses to Questions

**Key Review Criteria Required for Acceptance?**

**Methods**

-Are the objectives of the study clearly articulated with a clear testable hypothesis stated?

-Is the study design appropriate to address the stated objectives?

-Is the population clearly described and appropriate for the hypothesis being tested?

-Is the sample size sufficient to ensure adequate power to address the hypothesis being tested?

-Were correct statistical analysis used to support conclusions?

-Are there concerns about ethical or regulatory requirements being met?

Reviewer #1: The fungal isolates tested . It isn't clear how these were all speciated prior to testing. Was this carried out by molecular techniques ( please state how ) - This information should be included in the paper.

It is a pity that some mycetoma agents have not been used for comparison eg Scedosporium apiospermum as other non-pigmented fungi have been included. Was this an oversight ?

In the discussion the authors refer to the in silica assay. Can you expand on this ?

Reviewer #2: (No Response)

**Results**

-Does the analysis presented match the analysis plan?

-Are the results clearly and completely presented?

-Are the figures (Tables, Images) of sufficient quality for clarity?

Reviewer #1: Yes but see above 3rd comment

Reviewer #2: (No Response)

**Conclusions**

-Are the conclusions supported by the data presented?

-Are the limitations of analysis clearly described?

-Do the authors discuss how these data can be helpful to advance our understanding of the topic under study?

-Is public health relevance addressed?

Reviewer #1: Yes these are fine. My second point is relevant to limitations - albeit a minor one

Reviewer #2: (No Response)

**Editorial and Data Presentation Modifications?**

Reviewer #1: It would be helpful if the authors would say where they see the place of reference centres for mycetoma diagnosis. How many are there and do we need more ?

Reviewer #2: (No Response)

**Summary and General Comments**

Reviewer #1: See above

Reviewer #2: The manuscript by Lim et al describes the development of a novel PCR assay for the specific detection/identification of cases of eumycetoma caused by Madurella mycetomatis, based on a genome-wide comparative approach.

A series of potential gene targets for PCR were identified in putative coding regions throughout the M. mycetomatis genome, PCR primers were designed and tested against a panel of 60 M. mycetomatis isolates plus a range of sibling species in Madurella, other agents of eumycetoma, and a variety of less related filamentous fungi. 

One primer pair was selected due to its high specificity, and was shown to be slightly less sensitive that the previously described spscific PCR (which has since been shown to cross-react with another Madurella species.

A major question that should be addressed is whether this novel PCR is sufficiently sensitive to be used in diagnosis directly from clinical specimens eg fungal grains, or whether it will only be useful when working with pure cultures. I would encourage the authors to address this in the current manuscript as it would substantially strengthen the study.

The English usage is a little poor in places, and grammatical issue should be addressed on lines 35 (were not was); 50 (disappointing? is not clear); 54 (cause not causes); 57 (delete "a"); 57 (suggest: that we recommend"); sentence starting line 78 (developed is repeated)108 (using substituting?);127 (PCR reactions contained); 129 (were as follows); 132 (was this final stage really 10 seconds?); Table 2 (Madurella tropicana and spelling of Trematosphaeria); 154 Primer sets 4, 5 and 12 only amplified ...); 189 (delete "a")

PLOS authors have the option to publish the peer review history of their article (what does this mean?). If published, this will include your full peer review and any attached files.

Reviewer #1: Yes: Roderick J Hay

Reviewer #2: No
---

## [Editor Report · Decision Letter 1]

17 Oct 2020

Dear Mr. Lim,

We are pleased to inform you that your manuscript 'The development of a novel diagnostic PCR for Madurella mycetomatis using a comparative genome approach' has been provisionally accepted for publication in PLOS Neglected Tropical Diseases.

Best regards,

Husain Poonawala

Guest Editor

Todd Reynolds

Deputy Editor

---

## [Editor Report · Acceptance letter]

24 Nov 2020

Dear Mr. Lim,

We are delighted to inform you that your manuscript, "The development of a novel diagnostic PCR for *Madurella mycetomatis* using a comparative genome approach," has been formally accepted for publication in PLOS Neglected Tropical Diseases.

Best regards,

Shaden Kamhawi

co-Editor-in-Chief

Paul Brindley

co-Editor-in-Chief
